# Spin disorder control of topological spin texture

Hongrui Zhang [1,2,18] ✉, Yu-Tsun Shao [3,4,18], Xiang Chen [2,5,6,18] ✉,
Binhua Zhang [7,8,18], Tianye Wang [5], Fanhao Meng [1,2], Kun Xu [9],
Peter Meisenheimer [1], Xianzhe Chen [1,2], Xiaoxi Huang[1], Piush Behera[1],
Sajid Husain [1,2], Tiancong Zhu [2,5], Hao Pan [1], Yanli Jia[1], Nick Settineri[2],
Nathan Giles-Donovan[5], Zehao He [2,5], Andreas Scholl[10], Alpha N'Diaye [10],
Padraic Shafer [10], Archana Raja [11], Changsong Xu[7,8] ✉,
Lane W. Martin [1,2,12,13,14,15], Michael F. Crommie [2,5], Jie Yao [1,2],
Ziqiang Qiu [5], Arun Majumdar [9], Laurent Bellaiche [16], David A. Muller [4,17],
Robert J. Birgeneau [2,5] & Ramamoorthy Ramesh [1,2,5,12,14,15] ✉

Stabilization of topological spin textures in layered magnets has the potential to drive the development of advanced low-dimensional spintronics devices. However, achieving reliable and flexible manipulation of the topological spin textures beyond skyrmion in a two-dimensional magnet system remains challenging. Here, we demonstrate the introduction of magnetic iron atoms between the van der Waals gap of a layered magnet, $Fe_3GaTe_2$, to modify local anisotropic magnetic interactions. Consequently, we present direct observations of the order-disorder skyrmion lattices transition. In addition, non-trivial topological solitons, such as skyrmioniums and skyrmion bags, are realized at room temperature. Our work highlights the influence of random spin control of non-trivial topological spin textures.

Spin textures in magnetic materials result from the competition of magnetic exchange interaction, magnetic anisotropy, Dzyaloshinskii-Moriya interaction (DMI), and dipolar interaction[1-4]. The spin Hamiltonian in a perpendicular magnetic anisotropy (PMA) system can be approximated as:

$$H = -J\sum_{n.n.} \vec{s}_i \cdot \vec{s}_j - K_\perp \sum_i s_{iz}^2 + \sum_{n.n.} \vec{D}_{ij} \cdot \left(\vec{s}_i \times \vec{s}_j\right)$$
$$+ \Omega \sum_{<i,j>} \left( \frac{\vec{s}_i \cdot \vec{s}_j - 3\left(\vec{s}_i \cdot \hat{r}_{ij}\right)\left(\vec{s}_j \cdot \hat{r}_{ij}\right)}{\left|\vec{r}_{ij}\right|^3} \right) \quad (1)$$

where $J$ is the coefficient of exchange coupling, $K_\perp$ is the coefficient of perpendicular uniaxial anisotropy, $\vec{D}_{ij}$ is the local vector of DMI, $\Omega$ is the coefficient of magnetic dipole-dipole interaction. $\vec{r}_{ij}$ is the position vector from $i_{th}$ atom to $j_{th}$ atom, and $\hat{r}_{ij}$ is the unit vector along $\vec{r}_{ij}$. Spin textures such as skyrmions can be formed under a set of interaction parameters[1,2]. For example, through symmetry design in magnetic crystals or stacking configurations in magnetic multilayer films, DMI can be established, leading to the formation of ordered Bloch-type skyrmions[5,6], Néel-type skyrmions[7,8], and even antiskyrmions[9,10]. Compared to conventional skyrmions ($Q = \frac{1}{4\pi}\int M \cdot \left(\partial_x M \times \partial_y M\right)dxdy$, where $Q$ is the topological number and $M$ is the unit vector in the direction of the local magnetization, $Q = 1$), designing new types of topological spin textures ($Q \neq 1$) remains challenging with only current global parameter tuning. This limitation impedes the development of topological spin texture-based spintronic devices.

A full list of affiliations appears at the end of the paper. ✉e-mail: hongruizhang@berkeley.edu; chenx889@mail.sysu.edu.cn; csxu@fudan.edu.cn; rramesh@berkeley.edu

Theory and simulations suggest that the introduction of random disorder or frustration, leading to complex competition among various isotropic or anisotropic magnetic interactions with different energy scales, could potentially give rise to exotic topological spin textures beyond skyrmions[11–17]. As in relaxor ferroelectrics, the compositional inhomogeneity can be explicitly mapped into the three-dimensional (3D) Heisenberg model with cubic anisotropy in the presence of random electric fields[18]. This model is intrinsically unstable with the random local dipolar fields driving the system to breakup into nanodomains. Designing such nanodomains can give rise to extraordinary dielectric susceptibilities, energy storage, and piezoelectric performances[19–21]. These examples provide us with an illumination that the control of nano-magnetic domains may be achieved through the introduction of inhomogeneous spins, which can be employed to induce local anisotropic magnetic interaction in the system[22–24]. The Hamiltonian account for the contribution from the random spins can be written as:

$$\triangle H = \sum_i \vec{\phi}_i \cdot \vec{s}_i + \sum_i \psi_i \left( \vec{s}_i \cdot \hat{u}_i \right)^2 + \sum_{n.n.} \vec{D'}_{ij} \cdot \left( \vec{s}_i \times \vec{s}_j \right) \qquad (2)$$

where the first term is a linear random field term, and the second term is a second-order random anisotropy term. $\vec{\phi}_i$ is the on-site random field, $\hat{u}_i$ is the unit vector of on-site random uniaxial anisotropy. $\psi_i$ is the strength of the on-site random anisotropy. The third term is a second-order random DMI term. $D'_{ij}$ is the random local DMI vector. The introduction of these energy terms allows for random non-collinear spins, potentially favoring the stabilization of unique spin textures. Here, we demonstrate that the intercalation of spin-active species into a two-dimensional (2D) magnetic framework is a practical pathway to create an inhomogeneous spin distribution, which facilitates the coexistence of ordered/disordered magnetic domains and skyrmion lattices. Importantly, these intercalated random spins can assist the formation of rare topological solitons, such as skyrmioniums and skyrmion bags, even at room temperature.

## Results

### 2D magnets with intercalated random spins

$Fe_3GeTe_2$ is a well-known layered ferromagnet with a strong PMA[25,26] that hosts topological spin textures[27–32]. The intercalated iron atoms ($Fe^{int}$) between the vdW gaps were reported in this system, albeit the total iron concentration is typically lower than its stoichiometric number[32,33]. The $Fe_3GaTe_2$ compound was regarded to have a similar crystal structure but possesses a higher Curie temperature ($T_c$) than $Fe_3GeTe_2$[34,35]. Each unit cell of $Fe_3GaTe_2$ has AA' stacked two sublayers (Fig. 1a), which consists of a $Fe_3Ga$ layer sandwiched by two tellurium layers in each sublayer. The iron atoms within the sublayer occupy two inequivalent Wyckoff positions, labeled as $Fe^{top}$ ($Fe^{bot}$) and $Fe^{mid}$. $Fe^{int}$ is located at the octahedral intercalated sites within the vdW gaps. Here, we focus on the model system, $Fe_3GaTe_2$, and use Co-substituted $Fe_5GeTe_2$ without any measurable $Fe^{int}$ and $Fe_3GeTe_2$ with intercalated $Fe^{int}$ as references.

High-quality $Fe_3GaTe_2$ single crystals with several $Fe^{int}$ concentrations were synthesized via a self-flux method. (see Methods) The element ratios were investigated via energy-dispersive X-ray spectroscopy. The iron ratio in $Fe_3GaTe_2$ can exceed 3, and excess iron atoms are considered as $Fe^{int}$ atoms. The single-crystal X-ray diffraction (XRD) data determined the crystal structure of $Fe_3GaTe_2$ and revealed the existence of $Fe^{int}$ between the vdW gap. (Supplementary Fig. 1) Atomic-resolution, integrated differential phase contrast (iDPC)-scanning transmission electron microscopy (STEM) imaging was performed on cross-sectioned samples (Fig. 1b), directly confirming the atomic structure model obtained by the single-crystal XRD measurement. The $Fe^{int}$ atoms were randomly distributed within the vdW gap in $Fe_3GaTe_2$

with two different intercalation levels: 8.5% (Fig. 1c) and 65.3% $Fe^{int}$ (Fig. 1d). The peak intensities of $Fe^{top}$, $Fe^{bot}$, and $Fe^{int}$ in the iDPC-STEM image (Fig. 1c) are quantitatively analyzed. The intensity line profile of the $Fe^{bot}$ ($Fe^{top}$) site indicates its uniform distribution (green or blue line, Fig. 1e). On the contrary, the non-uniform and low-intensity line profile of the $Fe^{int}$ site (brown line, Fig. 1e) suggests a partial, random occupation of $Fe^{int}$. (Supplementary Fig. 2) Thus, both the single-crystal XRD and STEM measurements clearly demonstrate the existence of the randomly self-intercalated $Fe^{int}$ in $Fe_3GaTe_2$.

The random intercalation of $Fe^{int}$ pronouncedly affects the macroscopic magnetic properties in $Fe_3GaTe_2$, as evidenced by the macroscopic magnetization and magneto-transport measurements. Firstly, $Fe_3GaTe_2$ with $Fe^{int}$ manifests a ferromagnetic state with an enhanced $T_c$ above room temperature. (Supplementary Figs. 3–5) Further, the disordered spins induce a bifurcation between the zero-field cooling (ZFC) and field cooling (FC) magnetization-temperature (M-T) curves (Fig. 1f) with a spin-freezing temperature ($T_f$)[22]. As a comparison, faint bifurcation is observed in $Fe_{2.5}Co_{2.5}GeTe_2$ without $Fe^{int}$. (Supplementary Fig. 3) It is worth noting that both in-plane and out-of-plane M-T curves for the $Fe_3GaTe_2$ exhibit additional kink-like features at $T_f$ (Supplementary Fig. 3), which are likely the results of antiferromagnetically coupled, disordered spins. This picture is supported by the observation of a spin-flop transition and the exchange-bias behavior in the low-temperature magneto-transport measurements. (Supplementary Figs. 4,6) Interestingly, the virgin anomalous Hall curves for the $Fe_3GaTe_2$ with $Fe^{int}$ after ZFC lie outside the primary hysteresis loops (Fig. 1g and Supplementary Fig. 7). A stronger magnetic field is required to align the frozen antiferromagnetic coupling implies the existence of strong pining effect in the system. Lastly, the $T_c$ of $Fe_3GaTe_2$ with $Fe^{int}$ systems increases as the $Fe^{int}$ concentration increases (Fig. 1h), similar to the $Fe_3GeTe_2$ with $Fe^{int}$ system[33]; $T_f$ also follows the same trend. (Fig. 1i and Supplementary Fig. 7) Therefore, from the above structural and macroscopic magnetic characterization, $Fe^{int}$ can introduce disordered spins into the system and modify the magnetic couplings, altering the magnetic properties, such as the magnetic transition temperatures.

### Microscopic picture of disordered spins

To understand the nature of the disordered spins in $Fe_3GaTe_2$, we performed density-functional theory (DFT) calculations (see Methods) with a tunable level of $Fe^{int}$-site occupancy. We analyzed the magnetic couplings between the iron atoms and between sublayers separately. As expected, a strong ferromagnetic coupling is preferred among the nearest neighbor iron atoms within the $Fe_3Ga$ sublayer. (Supplementary Fig. 8) However, the third-nearest neighbor interaction ($Fe^{mid}$-$Fe^{mid}$ within the layer) favors an antiferromagnetic coupling with $J^{mm}$ = 4.3 meV in $Fe_3GaTe_2$ without $Fe^{int}$. Once $Fe^{int}$ is introduced, $J^{mm}$ increases to 7.93 meV. Furthermore, the $Fe^{int}$-$Fe^{int}$ interaction favors a stronger antiferromagnetic coupling with $J^{ii}$ = 17.3 meV. These antiferromagnetically coupled iron atoms are located in triangle lattices, inducing spin frustration. On the other hand, quantitatively, the ground-state energies of the representative magnetic states with out-of-plane spins, namely, interlayer ferromagnetic and anti-ferromagnetic coupled states, are calculated as a function of the $Fe^{int}$ concentration. The DFT calculation shows that the ferromagnetic order is preferred regardless of the $Fe^{int}$ concentration from 0% (i.e., $Fe_3GaTe_2$) to 100% (i.e., $Fe_4GaTe_2$) (Supplementary Fig. 8); however, at low concentration levels (e.g. 8.5% $Fe^{int}$ in $Fe_3GaTe_2$), the ground-state energies of the representative magnetic states are close and accessible due to the relatively small energy barriers (<10 meV/Fe) (Supplementary Fig. 8). Considering that the system cools down from above $T_c$, thermal fluctuations might be sufficient to overcome the energy barrier(s) to access the less-favored antiferromagnetic states. (Supplementary Note 1 and Supplementary Figs. 9–13) A corresponding Monte Carlo simulation supports the coexistence of antiferromagnetic and

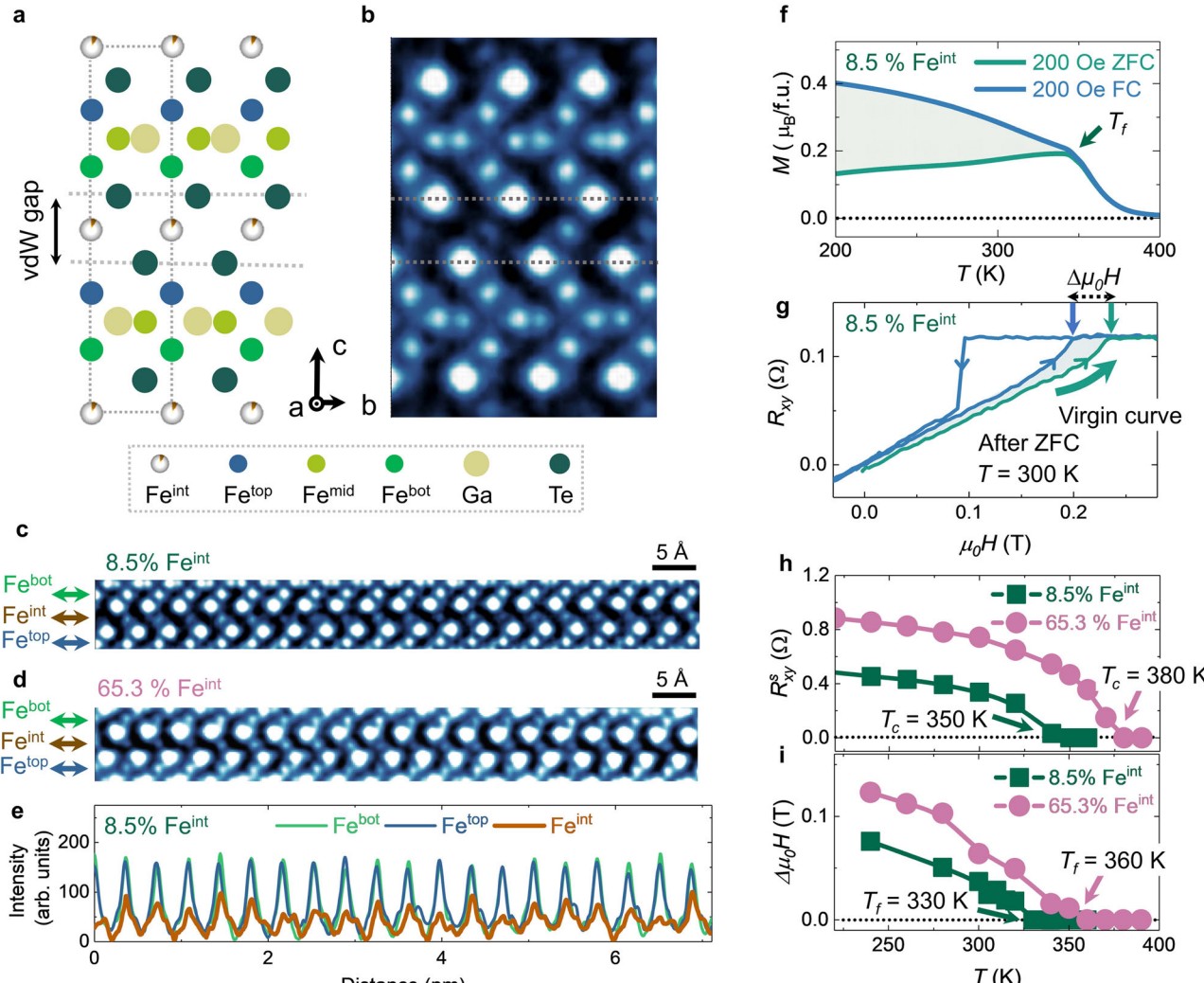

**Fig. 1 | Structural and magnetic characterization of Fe3GaTe2 with $Fe^{int}$. a** Side view of the atomic structural schematic image of $Fe_3GaTe_2$ with $Fe^{int}$. The $Fe^{int}$ and tellurium layers can be treated as a hexagonal $FeTe_2$-type structure. The $Fe^{int}$ atoms occupy the octahedral intercalated sites. **b** Atomic resolution cross-section iDPC-STEM image of $Fe_3GaTe_2$ with $Fe^{int}$. The atomic resolution cross-section iDPC-STEM image of vdWs gap for $Fe_3GaTe_2$ with 8.5% (**c**) and 65.3% (**d**) $Fe^{int}$. **e** Intensity line profiles of the $Fe^{bot}$, $Fe^{int}$, and $Fe^{top}$ atoms in Fig. 1c are shown in green, brown, and blue curves, respectively. The $Fe^{bot}$ and $Fe^{top}$ profiles are nearly uniform, while the $Fe^{int}$ is non-uniform. **f** M-T curves were measured after zero-field cooling (ZFC) and field cooling (FC). The dark green arrow marks the spin-freezing temperature ($T_f$). **g** The anomalous Hall curve for the $Fe_3GaTe_2$ nanoflake with 8.5% $Fe^{int}$ is measured at room temperature after ZFC. The dark blue curve refers to the virgin curve. Temperature dependence of saturated anomalous Hall resistance ($R^s_{xy}$, **h**) and delta saturated field ($\Delta\mu_0 H$, **i**) were measured for $Fe_3GaTe_2$ nanoflakes with 8.5% and 65.3% $Fe^{int}$ concentrations.

ferromagnetic phases below $T_f$. (Supplementary Fig. 14) Thus, the spins of iron can be antiferromagnetically coupled both between the sub-layers and between $Fe^{int}$ atoms below $T_f$, introducing random disorder and frustration to the system.

## Imaging the effect of disordered spins

One direct impact of the disordered spins of $Fe^{int}$ on the system is reflected on the magnetic domains, which can be imaged by conducting magnetic force microscopy (MFM) measurements (see Methods) on the $Fe_3GaTe_2$ nanoflakes with various $Fe^{int}$ concentrations at room temperature after ZFC. As a reference, $Fe_{2.5}Co_{2.5}GeTe_2$ nanoflake (with no $Fe^{int}$) exhibits stripe domains in these measurements (Fig. 2a), with an approximately uniform wavevector of the magnetic modulation ($q$-vector). In contrast, disordered magnetic domains are observed in $Fe_3GaTe_2$ with the introduction of $Fe^{int}$. Specifically, in $Fe_3GaTe_2$ nanoflakes with 5.0% (Fig. 2b) and 8.5% (Fig. 2c and Supplementary Fig. 15) $Fe^{int}$, numerous dislocations are surrounded by stripes with different $q$-vector directions. As the $Fe^{int}$ concentration reaches up

to 65.3% $Fe^{int}$ (Fig. 2d and Supplementary Fig. 15), the domain pattern exhibits a complex labyrinthine domain without any stripe domains. These results indicate that the disorder spins can directly cause increased magnetic domain chaos. Such experimental observation can be closely reproduced by the micromagnetic simulation (see Fig. 2e−g, supplementary Fig. 16 and Methods). The random magnetic aniso-tropy of various densities is introduced into the system, and then the evolution of the magnetic domain is simulated as the temperature cools down from near $T_c$, exactly following the experimental process. As the fraction of defects increases, the magnetic domain pattern transitions from stripe domains (Fig. 2e) to complex labyrinthine domains (Fig. 2g). At a defect density of 20%, the system exhibits the formation of various domain patterns, including bubble, target, ring-shaped, and net-shaped domains (Fig. 2g).

Noticeably, there is another feature in the $Fe^{int}$ concentration evolution of the domain structure, namely, the MFM contrast in $Fe_3GaTe_2$ with lower $Fe^{int}$ concentrations is non-uniform. In compar-ison, a uniform contrast is recorded in the nanoflakes with 65.3% $Fe^{int}$

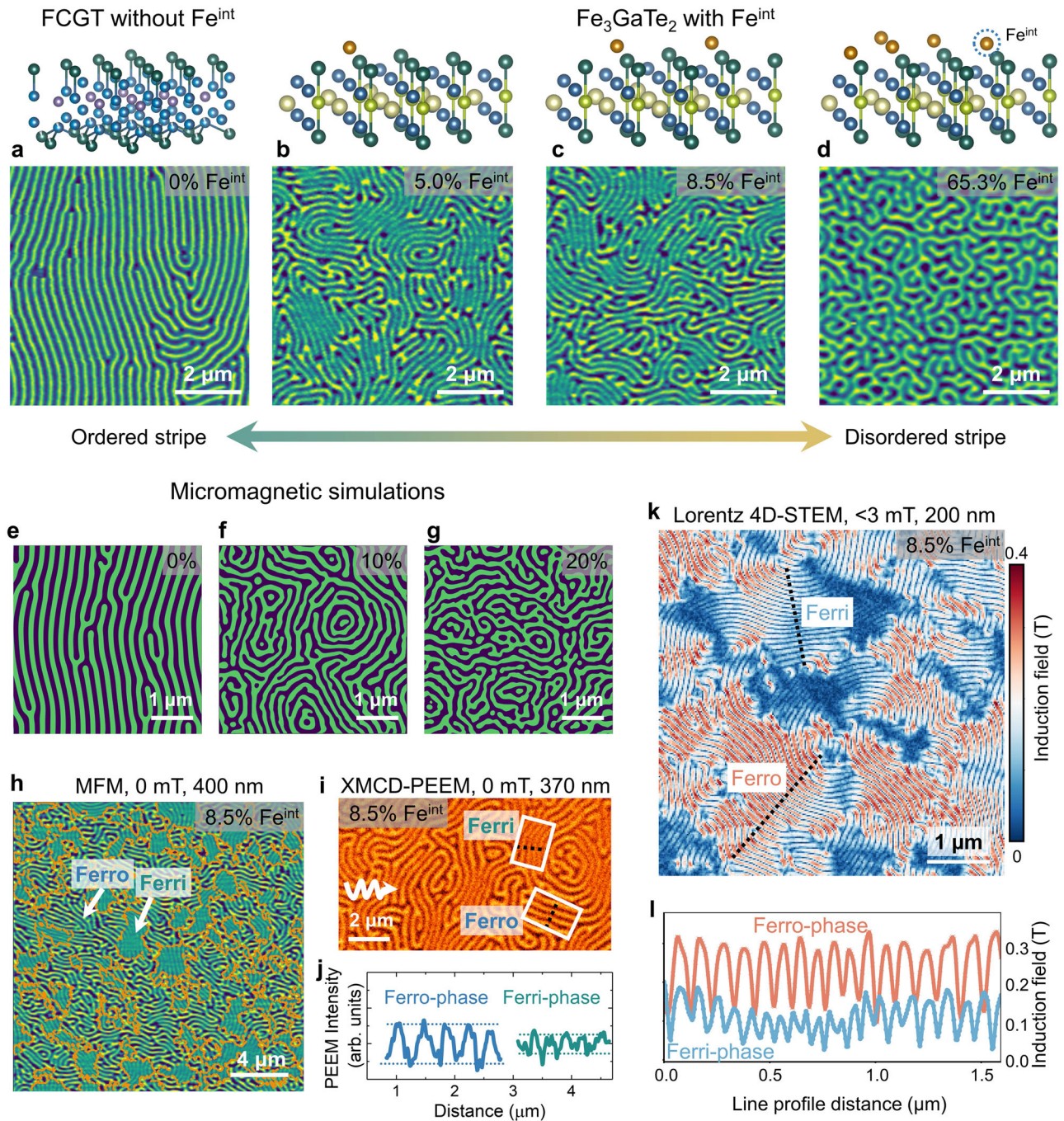

**Fig. 2 | Imaging the effect of disordered spins at room temperature.** The MFM images of the Fe$_{2.5}$Co$_{2.5}$GeTe$_2$ without Fe$^{int}$ (**a**) and Fe$_3$GaTe$_2$ with various Fe$^{int}$ concentrations (**b**, 5.0%, **c**, 8.5%, and **d**, 65.3%) nanoflakes were measured at room temperature after ZFC. The thickness of the nanoflakes here is 200 - 400 nm. The up panels display the schematic images of Fe$^{int}$ concentrations. **e**–**g** Simulated magnetic domain images as the density of random anisotropy increases. (Supplementary Fig. 16). **h** The MFM image of a 400-nm-thick nanoflake at room temperature under zero field shows two different regions with strong (Ferro-phase) and weak (Ferri-phase) frequency contrasts, demonstrating the micrometer-scale phase separation. The orange lines denote the phase boundaries between the Ferro-phase and Ferri-phase. **i** The XMCD-PEEM image of the magnetic domains for a 370 nm thick flake at room temperature. The white arrow represents the projection of the 60° off-normal incident direction of the X-ray. **j** The line profiles across two-phase regions in the PEEM image in the white box in Fig. 2i. **k** The 4D-LSTEM mapping of a 200-nm-thick Fe$_3$GaTe$_2$ with 8.5% Fe$^{int}$ nanoflake shows the details of the induction field around the spin textures collected at room temperature under zero field. **l** Line profiles of the two-phase regions in the 4D-LSTEM image along the black dotted lines in Fig. 2k. The induction fields of the Ferro-phase (dark red) and Ferri-phase (light blue) regions are about 0.3 T and 0.15 T, respectively. Notably, since the 4D-LSTEM technique is only sensitive to the domain walls, the periodicity of the line profiles is doubled compared to that of the MFM and XMCD-PEEM images.

concentration. For example, the non-uniform contrast is more apparent in the MFM image of a 400 nm nanoflake of Fe$_3$GaTe$_2$ with 8.5% Fe$^{int}$ (Fig. 2h), where there is an intricate pattern consisting of two distinctly contrasting regions: a stripe domain region with weaker contrast and a brighter domain region with numerous magnetic stripe dislocations. The antiferromagnetic domains that can be accessible in the lower Fe$^{int}$ concentration system based on the DFT calculations do not contribute to the intensity in MFM images. Consequently, the weak contrast regions (Fig. 2h) can be assumed to be the consequence of the coexistence of interlayer antiferromagnetism with a ferromagnetic

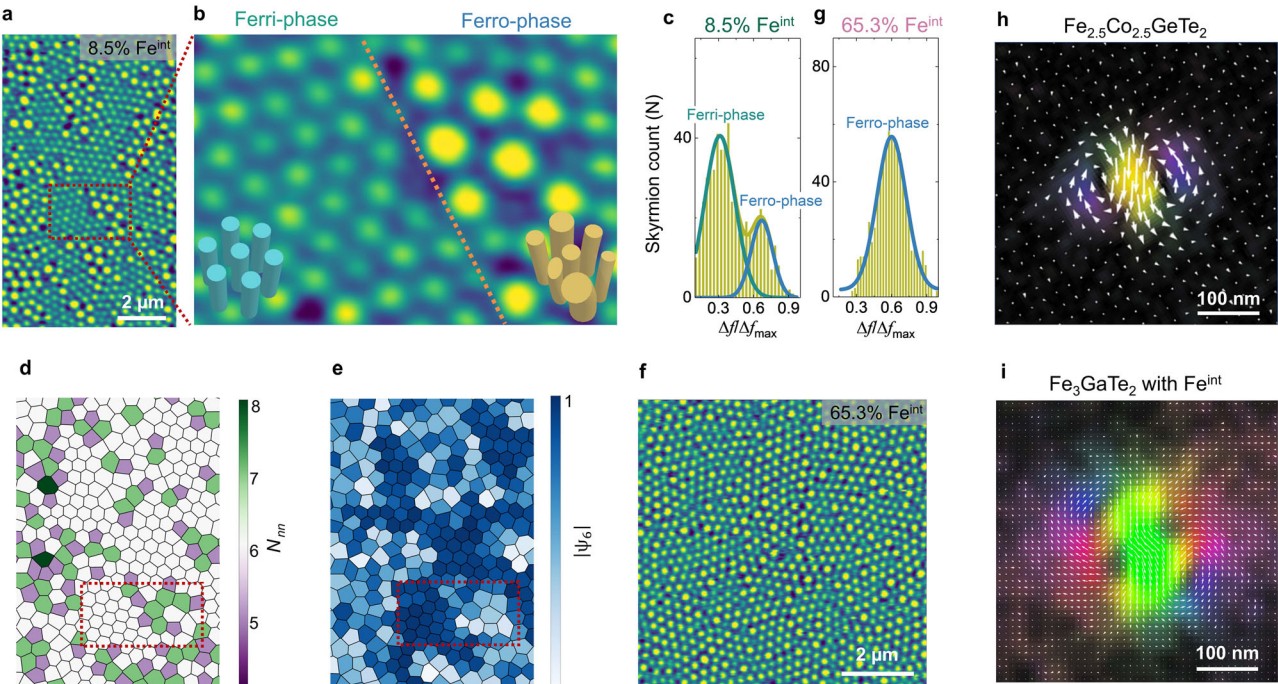

**Fig. 3 | Skyrmion ordering. a** The MFM image of a Fe$_3$GaTe$_2$ with 8.5% Fe$^{int}$ nanoflake at room temperature and zero field exhibits two distinct skyrmion lattices. **b** Zoom in on the dark red box in Fig. 3a. The orange dotted line denotes the skyrmionic phase boundary. The contrast frequency shift of the disordered skyrmion lattice (Ferro-phase) is typically higher than that of the ordered skyrmions (Ferri-phase). Also, the disordered skyrmion size is non-uniform compared to the relatively uniform size of the ordered skyrmions. **c** Statistical histogram of skyrmion counts as a function of the contrast frequency shift in Fig. 3a. The dark yellow curve is the overall fit of the Gaussian function in Fig. 3c. The high-frequency contrast range (blue curve) refers to the Ferro-phase, while the low-frequency contrast range (dark green curve) corresponds to the Ferri-phase. **d, e** Nearest neighbor ($N_{nn}$) and bond orientational ($|\psi_6|$) maps of the zoom in Fig. 3a. Statistical

analysis was conducted following the same methodology as in the previous reference[40]. **f** The MFM image of a Fe$_3$GaTe$_2$ with 65.3% Fe$^{int}$ nanoflake at room temperature and zero field shows a disordered skyrmion lattice. **g** Statistical histogram of skyrmion counts as a function of the contrast frequency shift in Fig. 3f. Only one peak fits the curve well, indicating single-phase skyrmions. **h** The induction field mapping of a typical ordered Néel-type skyrmion, observed in Fe$_{2.5}$Co$_{2.5}$GeTe$_2$ (190 nm) with a global breaking inversion symmetry at room temperature, is composed of clockwise and counterclockwise spin curl. **i** The induction field mapping of a disordered skyrmion in a 200-nm-thick Fe$_3$GaTe$_2$ with 8.5% Fe$^{int}$ nanoflake at room temperature displays a more complex feature, likely reflecting a 3D spin texture.

background (labeled as Ferri-phase), while the strong contrast regions (Fig. 2h) correspond to the predominant ferromagnetic domains without antiferromagnetically coupled spin domains (labeled as Ferro-phase).

A similar pattern with distinct contrast is also observed in surface sensitive (~5 nm) X-Ray Magnetic Circular Dichroism-photoemission electron microscopy (XMCD-PEEM, see Methods) image for a Fe$_3$GaTe$_2$ with 8.5% Fe$^{int}$ nanoflake (Fig. 2i). Under identical imaging conditions, the intensity of the Ferro-phase is about 2X higher than that of the Ferri-phase, as presented in the line profile in Fig. 2j. To further quantitatively map the magnetic induction fields of two phases in Fe$_3$GaTe$_2$ with 8.5% Fe$^{int}$, we employed four-dimensional (4D) Lorentz-STEM (LSTEM) coupled with an electron microscopy pixel array detector (EMPAD; see Methods)[36,37]. The magnetic induction field can be derived by quantitatively measuring the deflection angles of the electron beam in each diffraction pattern. The results show the absolute magnitude of the magnetic induction fields (Fig. 2k), which clearly exhibit two regions of distinct contrast. The induction fields around the spin textures in the Ferro- and Ferri-phases (Fig. 2l) are ~ 0.3 T and ~ 0.15 T, respectively, as guided by the line profiles along the black dotted lines (Fig. 2k). It is worth mentioning that similar behavior is also observed in Fe$_3$GaTe$_2$ nanoflakes with 6.7% Fe$^{int}$. (Supplementary Fig. 17) Thus, both bulk- and surface-sensitive probes, including MFM, XMCD-PEEM, and 4D-LSTEM real-space imaging techniques, directly corroborate the coexistence of two-phase magnetic domains in Fe$_3$GaTe$_2$ nanoflakes with a lightly intercalated Fe$^{int}$.

## Formation of ordered/disordered skyrmion lattices

Having established the formation of different magnetic domains through random spins, we further explored the stabilization of topological skyrmion lattices based on the order/disorder magnetic domain. Zero-field skyrmions in nanoflakes were stabilized at room temperature using the local stray fields generated by the MFM tip[38,39].(Supplementary Fig. 18) A triangle-ordered Néel-type skyrmion lattice can be stabilized in the Fe$_{2.5}$Co$_{2.5}$GeTe$_2$ nanoflake[40]. Interestingly, a distinct contrast of the skyrmion lattices in the Fe$_3$GaTe$_2$ nanoflakes with 8.5% Fe$^{int}$ is present (Fig. 3a,b), corresponding to magnetic domain-phase separation (Fig. 2h). Due to the weaker stray field of the Ferri-phase, its skyrmions are smaller than those of the Ferro-phase[3]. We present a statistical analysis of the MFM image (Fig. 3a) wherein two distinct regimes are identified by carrying out a Gaussian fitting of the image intensity distribution (Fig. 3c), corresponding to the Ferri- and Ferro-phase skyrmions. In the Ferri-phase regions, the number of nearest neighbors ($N_{nn}$) is approximately 6 in Fig. 3d, and the bond orientational parameter ($|\psi_6|$) is close to 1 in Fig. 3e, indicating a solid-phase skyrmion lattice[40]. In contrast, the presence of numerous 5-7 pairs caused by the stripe dislocations and the $|\psi_6|$ parameter significantly below 1 suggests a liquid-phase skyrmion lattice in the Ferro-phase regions. Similarly, a single Ferro-phase skyrmion lattice in the Fe$_3$GaTe$_2$ nanoflake with 65.3% Fe$^{int}$ (Fig. 3f), confirmed by Gaussian fitting of the image intensity distribution (Fig. 3g), shows no long-range ordering of skyrmions.

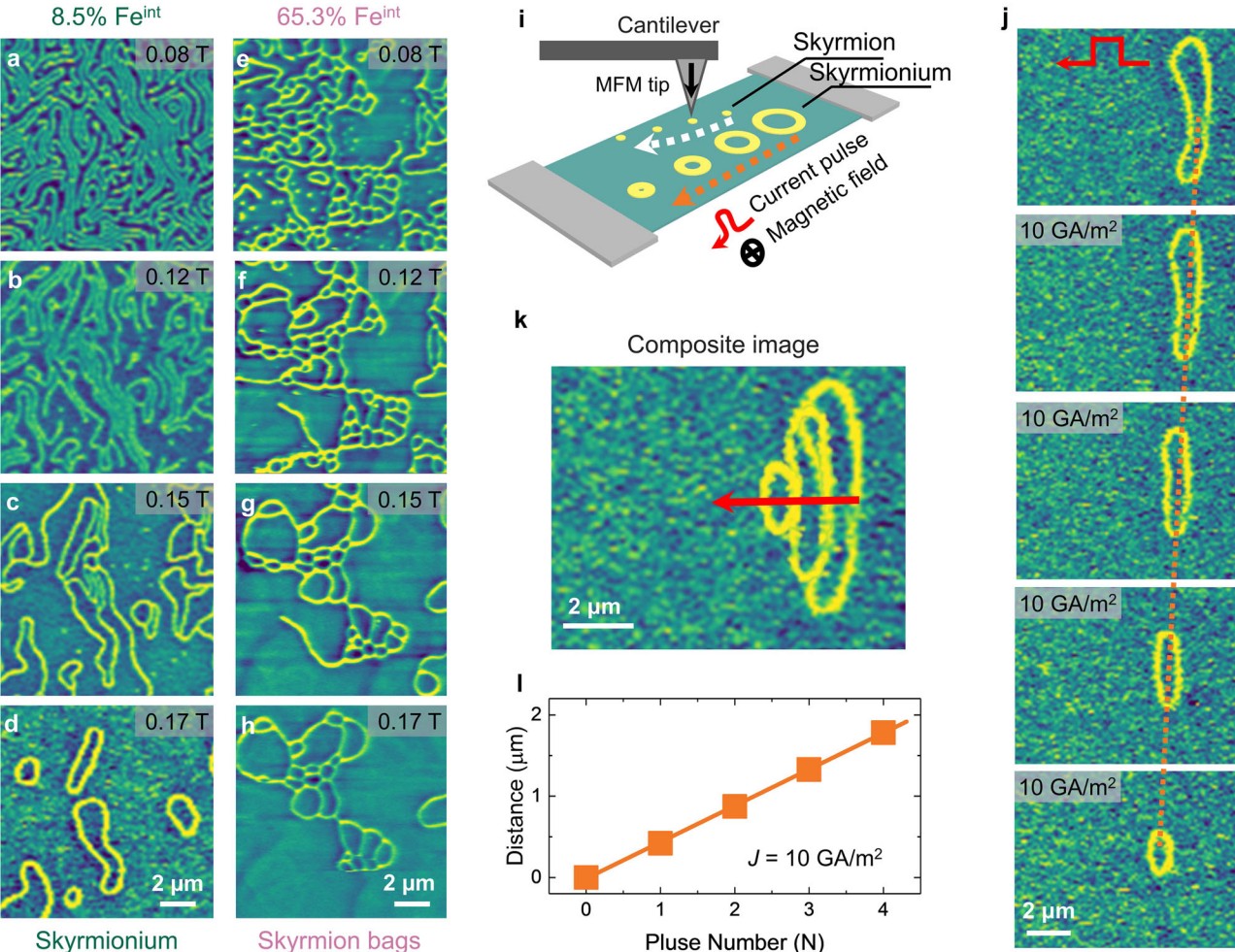

**Fig. 4 | Manipulation of topological number and skyrmionium dynamics.** The magnetic field dependence of MFM images for the Fe$_3$GaTe$_2$ nanoflakes with 8.5% Fe$^{int}$ (**a**–**d**) and 65.3% Fe$^{int}$ (**e**–**h**) were obtained at room temperature. The MFM contrast in Fig. **a** becomes uniform under the magnetic field in panels **b**–**d**, indicating that the antiferromagnetic domains transition into the ferromagnetic domains. **i**, Schematics of current-induced skyrmion and skyrmionium motion. **j** Sequential MFM images showing skyrmionium displacement after injecting 1 current pulse with 10 GA/m$^2$ amplitude. The orange dashed lines are included as visual guides of skyrmionium motion. **k** Skyrmionium trajectories of the current-induced motion. The red arrows refer to the motion direction, which is along the current pulse direction. **l** The current pulse-dependent motion distance indicates the velocity of the skyrmionium is ~0.42 mm/s at room temperature under a current density of 10 GA/m$^2$.

The character of the domain wall in the Fe$_3$GaTe$_2$ with Fe$^{int}$ system remains of the Néel-type (Supplementary Fig. 19), similar to the Fe$_{2.5}$Co$_{2.5}$GeTe$_2$ without Fe$^{int}$[39,41]. The skyrmion contrast in the LSTEM/MFM image cannot distinguish any difference between Fe$_{2.5}$Co$_{2.5}$GeTe$_2$ and Fe$_3$GaTe$_2$ nanoflakes with Fe$^{int}$. To investigate the impact of random spins on the ordering of the skyrmion lattice, the magnetic induction fields of these skyrmions were studied using 4D-LSTEM coupled with an EMPAD. (see Methods) The induction field around a model Néel-type skyrmion tube in Fe$_{2.5}$Co$_{2.5}$GeTe$_2$ with a broken crystallographic inversion symmetry is composed of both clockwise and counterclockwise curls (Fig. 3h). Remarkably, the detailed curling of the induction field around a Ferro-phase skyrmion in Fe$_3$GaTe$_2$ with 8.5% Fe$^{int}$ (Fig. 3i) exhibits more complex features. It does not resemble that of a simple 2D Néel-type skyrmion tube structure or a skyrmion with a higher topological number as theoretically predicted in Fe$_3$GeTe$_2$[42] and other frustrated system[14,43]. This discrepancy is resolved if the disordered skyrmion observed is considered to be a 3D twisted/bent spin texture[44–46]. Thus, the random repulsive interactions among non-uniform distorted skyrmions disrupt the densest hexagonal packing, resulting in a disordered skyrmion lattice.

## Stabilization of skyrmionium and skyrmion 'bags'

Applying a global magnetic field can further amplify the effect of the disordered spins on the system, as monitored by the magnetic field-dependent MFM measurements conducted at room temperature in the Fe$_3$GaTe$_2$ nanoflake with 8.5% Fe$^{int}$ (Fig. 4a–d) and 65.3% Fe$^{int}$ (Fig. 4e–h). As the out-of-plane magnetic field increases, in the Fe$_3$GaTe$_2$ nanoflake 8.5% Fe$^{int}$, the contrast of the MFM image becomes uniform, as depicted in Fig. 4a, b and Supplementary Fig. 20. Then, the stripe domains reverse one by one due to the strong perpendicular anisotropy energy [Supplementary Fig. 3, $K_u$ ($T$ = 300 K) = 4.0 ×10$^5$ J/m$^3$] of Fe$_3$GaTe$_2$ at room temperature. The Fe$_{2.5}$Co$_{2.5}$GeTe$_2$ system [$K_u$ ($T$ = 300 K) = 2.4 ×10$^5$ J/m$^3$, $K_u(T)$ ~ $M^3{}_S(T)$] also shows similar behavior at low temperatures[39]. Intriguingly, the ring-shaped dislocations, composed of two opposite directions of edge dislocations, transform into isolated skyrmioniums ($Q$ = 0) (Fig. 4d). The size of the skyrmioniums is on the order of micrometers. Finally, the skyrmioniums shrink, collapse, and then transition into a ferromagnetic state. Other edge dislocations and short stripe domains either shrink to skyrmions ($Q$ = ±1) or disappear (Fig. 4c, d). In the Fe$_3$GaTe$_2$ nanoflake with 65.3% Fe$^{int}$, high-density edge dislocations intersect to stabilize a net-shaped domain instead

of a ring-shaped domain (Fig. 4e). Such net-shaped domain can be regarded as the composition of an outer skyrmion and an arbitrary number of inner skyrmions, i.e., skyrmion 'bags'. The number of the inner skyrmions ($S$) determines the topological number [$Q = \pm (S-1)$]. As the magnetic field increases, some of the inner skyrmions melt, resulting in a reduction of the topological number. When the magnetic field reaches the saturation field, the system transitions into a single-domain state. Thus, in $Fe_3GaTe_2$ with $Fe^{int}$ system, the density of edge dislocations fundamentally determines the formation of isolated skyrmion, skyrmionium, or skyrmion 'bags', leading to the manipulation of various topological numbers. Under a moderate magnetic field, due to the strong pinning effect facilitated by the introduced random disorder, these non-trivial topological spin textures are more likely to exist and survive. In comparison, in the $Fe_{2.5}Co_{2.5}GeTe_2$ without $Fe^{int}$ system, only a few isolated skyrmions survive at low temperatures in the initial magnetization process[39], possibly originating from the smaller pinning effect from some structural defects/disorder.

## Skyrmionium dynamics

The non-trivial topological spin textures, such as skyrmioniums, have their unique characteristics and the potential for racetrack memory applications[47–52]. It is well known that the lateral motion of a magnetic skyrmion driven by pulse current (Supplementary Fig. 21), caused by the skyrmion Hall effect[53–55], imposes severe limitations on the practical use in racetrack memory applications[56]. Encouragingly, skyrmionium can move without the skyrmion Hall effect by the pulse current due to the opposite Magnus forces acting on the skyrmion components with $Q = +1$ and $Q = -1$. To verify this, we performed the current-induced skyrmionium motion at room temperature. The experimental image is illustrated (Fig. 4i). An external magnetic field (-0.194 T) was applied to a 195 nm thick nanoflake to stabilize an individual skyrmionium (Fig. 4j). Each image was acquired after injecting one current pulse with a current density of 10 GA/m$^2$ and a duration of 1 ms. As expected, the skyrmionium measurably moves forward without deflection (Fig. 4j, k). Meanwhile, it shrinks as the pulse number increases, possibly due to the decrease in domain wall energy caused by thermal or spin-orbit torque effects. The threshold current density for the skyrmionium motion is smaller than that of the skyrmion (Supplementary Fig. 21), aligning with the findings of the micromagnetic simulation[15,57]. The distance moved after each pulse current is uniform (Fig. 4l). The calculated velocity of the skyrmionium at room temperature is ~0.42 mm/s under a current density of 10 GA/m$^2$.

## Discussion

As in classical ferromagnets, the domain pattern arises from the competition between short-range magnetic and long-range dipolar interactions[1,3]. The intercalated magnetic $Fe^{int}$ can introduce random local magnetic interactions, e.g., the pinning effect, which was simulated by a random local magnetic anisotropy. (Fig. 2e–g and Supplementary Fig. 16) These intercalated spins can result in stripe dislocations and labyrinthine domains as the temperature decreases from $T_c$. In the Ferro-phase state, the number of stripe dislocations increases with the concentration of $Fe^{int}$. However, in the Ferri-phase state, due to the lower magnetic dipole-dipole interaction, the domain wall energy dominates, and it remains stripe domains. (Supplementary Note 2).

Although the crystal structure of $Fe_3GaTe_2$ with/without $Fe^{int}$ has a centrosymmetric space group ($P6_3/mmc$) and, thus, in principle, should not exhibit a global DMI. (Supplementary Fig. 22) The existence of a surface oxidized layer[29] or inhomogeneous $Fe^{mid}$ [32] might lead to a global DMI, resulting in a Néel-type character of the domain wall. In addition, the random disorder through intercalation in vdW magnets promotes the formation of phase-separated magnetic domains. By applying the stray field of MFM tips, the stripe domain region forms an ordered skyrmion lattice, and the labyrinthine domain region forms a disordered skyrmion lattice. Furthermore, the ring- or net-shaped domains host a lower barrier energy due to the strong pinning effect, which can be regarded as the seed of the various intriguing topological spin textures. The strong PMA of $Fe_3GaTe_2$ at room temperature allows the stripe domain to reverse rather than breaking into bubbles under a magnetic field. Thus, pinned ring- or net-shaped domains prefer to survive and form isolated skyrmions, skyrmioniums, or skyrmion bags. Ultimately, the density of edge dislocation induced by the random spins in the system determines the nature of topological spin textures.

Our study establishes the role of disordered spins induced by the intercalated iron atoms in controlling the spin textures in a layered ferromagnet $Fe_3GaTe_2$. Such random spin disorder enables the tuning of local magnetic interactions, allowing the manipulation of the order of the skyrmion lattice at a macro scale, even at room temperature. More strikingly, various unusual topological spin textures, including 3D distorted skyrmions, skyrmioniums, and skyrmion 'bags' can be realized by varying the level of intercalation. Our work highlights the significant impact of random spins residing in the vdW gaps in shaping unconventional topological spin textures. This method is likely promising for broad applications in other vdW magnets as well.

## Methods

### Sample synthesis

Single crystals of $Fe_3GaTe_2$ were grown by the self-flux method. Starting materials comprised of elemental iron granules (99.99%), gallium chunks (99.99%), and tellurium shots (99.999%) with a nominal molar ratio of 1: 1: 2 were fully mixed together inside the glovebox. The starting mixture was then evacuated and sealed inside a quartz tube. The sealed quartz tube was positioned horizontally inside a muffle furnace during the growth process. The reaction temperature was maintained at 900 °C under isothermal conditions for a duration of 6 days. The single crystals were obtained by quenching the furnace at 750 °C. Higher self-intercalated $Fe_3GaTe_2$ crystals were obtained by a slightly different growth method: a nominal molar ratio of iron: gallium: tellurium = 3: 1: 2 were fully mixed; then the raw materials were subjected to a higher growth temperature at 1000 °C, which were later slowly cooled down to 900 °C, before quenching to room temperature in the last step of synthesis.

### Energy-dispersive X-ray spectroscopy (EDS)

A Quanta 3D field emission gun (FEG) scanning electron microscope (SEM) was used in this research. Energy dispersive spectroscopy (EDS) was carried out on multiple single crystals of $Fe_3GaTe_2$ mounted with carbon tape using an Oxford EDS attached to the SEM. The atomic ratio of iron, gallium, and tellurium from the multiple sites of each sample is consistent and averaged as 3.08(2): 0.96(2): 2, which is very close to the 3:1:2 stoichiometry. On the other hand, the EDS measurements on the flakes from the high self-intercalated crystals, which display strong ferromagnetism from the MFM measurements, indicate a higher self-intercalated Fe level with 3.65(8): 1: 1.92(4). In $Fe_3GaTe_2$ with higher $Fe^{int}$ concentrations, there exist tellurium and gallium vacancies, which could lead to a slight overestimation of $Fe^{int}$ concentration.

### Single crystal X-ray diffraction (XRD)

Single crystal XRD measurements of the $Fe_3GaTe_2$ samples were carried out using the same conditions as outlined in the reference[41]. A solution was found using ShelXT2 in space group P6$_3$/mmc (No. 194) with lattice parameters $a = b = 4.080$ Å, $c = 16.138$ Å, $\alpha = \beta = 90°$, $\gamma = 120°$. Although the space group is the same as that of either Fe$_{3-x}$GeTe$_2$ or Ni$_{3-x}$GeTe$_2$, the solution indicates that a small percentage of iron (labeled as $Fe^{int}$ site) is inserted between the van der Waals layers, similar to the case of Ni$_{3-x}$GeTe$_2$[58]. During the initial refinements of the data, the site occupancy of all atoms was free to vary. It was found that the Te and Fe$^{top(bot)}$ sites were very stable and close to 1, while the site occupancy of gallium was slightly less than 1 and close to 0.96. After fixing the site occupancy

of tellurium, $Fe^{top(bot)}$, and gallium to 1, 1, and 0.96, respectively, further refinement shows the site occupancy of $Fe^{mid}$ (within the same horizontal plane as gallium) and $Fe^{int}$ (inserted between the van der Waals layers) were 0.925 and 0.085, respectively. Therefore, from the refinement of the data, the best solution of the atomic ratio of iron, gallium, and tellurium equals 3.01: 0.96: 2, which is close to the elemental composition observed by the EDS measurements.

## (Lorentz) scanning transmission electron microscopy

Cross-section TEM specimens were prepared from the $Fe_3GaTe_2$ nanoflakes using a Thermo Fisher Helios G4 UX focused ion beam. The preparation involved an initial milling with a $Ga^+$ ion beam of 30 and 5 keV, followed by a final polishing step at 2 keV to minimize ion beam damage. Carbon and platinum protective layers were deposited before milling to protect the surface. Simultaneous HAADF- and iDPC-STEM were acquired by using a Cs-corrected Thermo Fisher Scientific "Kraken" Spectra 300 operated at 300 keV, with a probe semi-convergence angle of 30 mrad, and a beam current of 15 pA. The intensity line profiles across the Te-$Fe^{int}$-Te atomic planes were obtained by identifying the atomic column positions with Gaussian 2D peak fitting using a custom Python script and the Atomap package[59]. Four-dimensional (4D) Lorentz scanning transmission electron microscopy (LSTEM) experiments were conducted under the same conditions as described in the references[39,41].

## Magnetization measurements

Magnetization of the single crystals was carried out with a superconducting quantum interference device magnetometer (Quantum Design, 2-400 K, 7 T), with the magnetic fields applied along both the out-of-plane and in-plane directions of the crystal.

## Transport measurements

Electronic transport measurements were performed in a Cryogen Free Measurement System from Cryogenic Ltd., using a Keithley 2400 source and 2182 nanometer. The applied current was fixed to 100 μA.

## DFT Calculations

DFT calculations were performed using the Vienna Ab initio Simulation Package (VASP) with the projector augmented wave (PAW) method. The local density approximation to the exchange-correlation functional without a Hubbard U correction was employed; this has previously been shown to describe the structural properties of isomorphic $Fe_3GeTe_2$ well. The plane wave energy cutoff was set at 400 eV. A vacuum layer of $\approx 20$ Å was adopted to avoid the interaction between periodic images. The Hellmann–Feynman forces were taken to be converged when they became smaller than 0.001 eV Å$^{-1}$ on each ion. The Brillouin zone was sampled by a $15 \times 15 \times 1$ k-point mesh for the monolayer unit cell. The four-state energy mapping method was performed to obtain magnetic parameters from DFT total energies, for which the $3 \times 3 \times 1$ supercell and $5 \times 5 \times 1$ mesh were adopted. In calculating DMI and single-ion anisotropy (SIA), the spin-orbit coupling was included. Our Monte Carlo (MC) simulations were performed using the calculated magnetic exchange interactions. The $100 \times 100 \times 1$ supercells were adopted in the study. For each configuration, 10,000 and 100 00 MC steps per site were performed for equilibrating the system and statistical averaging, respectively.

## MFM measurements

The MFM images of $Fe_3GaTe_2$ flakes on a $SiO_2$/Si wafer were measured at room temperature using an Asylum Research MFP-3D Origin$^+$ scanning probe microscope. The zero-field skyrmion lattices were induced by an MFM tip with a strong stray field. A low stray field of MFM tips was used to measure the MFM image to avoid magnetic interactions of the tip with the sample or an applied field. The spatial magnetic resolution was better than 25 nm. We used a two-step method in frequency-modulation mode to measure the MFM images. The distance between the sample surface and the MFM tip was fixed at 50 nm.

## XMCD and PEEM

XMCD measurements were conducted at room temperature at Beamline 6.3.1 of the Advanced Light Source by alternating the magnetization parallel and antiparallel to the direction of circularly polarized X-rays at normal incidence. PEEM images were obtained at room temperature at Beamline 11.0.1 of the Advanced Light Source using left- and right-circular polarized X-rays with 60° off-normal incidence and a photon energy of 706.8 eV, corresponding to the Fe $L_3$ edge. The circularly polarized X-ray is incident at 60° off-normal angle onto the sample surface from the left of Fig. 2f, which has contributions from the in-plane component of the magnetization. The uniform contrast among stripe regions along different directions indicates little to no in-plane magnetization component in the domains.

## Micromagnetic simulation

Micromagnetic simulation of the labyrinth and stripe domains was performed using the open-source software MuMax3[60]. To simulate the nucleation and evolution of magnetic domains of $Fe_3GaTe_2$ as the temperature decreases from near $T_c$ in each simulation, we set the saturation magnetization $M_s$ to increase step by step from 0.05 to 1.0 of the maximum value $3.76 \times 10^5$ $A/m$, and let the system evolve (relax) until stable at each step. The magnetizations were initialized with a randomized state before the first step in each simulation. The phenomenological power law of the dependence of magnetic parameters on the saturation magnetization are set as follows: $A(m_s) = A_0 m_s^2, K_u(m_s) = K_{u0} m_s^3, D(m_s) = D_0 m_s^2$, where $m_s$ is the magnetization ratio coefficient from 0.05 to 1.0 with steps of 0.05, $A$ is the exchange stiffness, $K_u$ is the perpendicular uniaxial magnetic anisotropy energy constant, and $D$ is the DMI strength. The cubic power of $m_s$ in $K_u$ is to represent the lower magnetic anisotropy energy at high temperatures compared to magnetic dipole-dipole interaction, which is proportional to $m_s^2$. Different densities of defects are simulated by random 50 nm-sized grains with larger magnetic anisotropy to simulate the local pinning effect. Magnetic parameters of the defect-free regions are $A_0 = 7.5 \times 10^{-12} J/m, D_0 = 0.6 mJ/m^2, K_{u0} = 2 \times 10^5 J/m^3$, and the defect regions host a three times of magnetic anisotropy $K_{u0}$. The simulations were run with a range of defect densities from 0% to 20%.

## Reporting summary

Further information on research design is available in the Nature Portfolio Reporting Summary linked to this article.

## Data availability

The data that support the figures and other findings of this study are available from the corresponding authors upon reasonable request.

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

## Acknowledgements
Work at Lawrence Berkeley National Laboratory was primarily supported by the U.S. Department of Energy, Office of Science, Office of Basic Energy Sciences, Materials Sciences and Engineering Division under Contract No. DE-AC02-05CH11231 within the Quantum Materials Program (No. KC2202) and U.S. Department of Energy, Office of Science, Office of Basic Energy Sciences, Materials Sciences and Engineering Division under Contract No. DE-AC02-05-CH11231 (Codesign of Ultra-Low-Voltage Beyond CMOS Microelectronics (MicroelecLBLRamesh)) for the development of materials for low-power microelectronics. H.Z. and R.R. acknowledge the Air Force Office of Scientific Research 2D Materials and Devices Research program through Clarkson Aerospace Corp under Grant No. FA9550-21-1-0460. Y.T.S. and D.A.M acknowledge financial support from the Department of Defense, Air Force Office of Scientific Research under Grant No. FA9550-18-1-0480. Z.Q. and T.W. acknowledge the U.S. Department of Energy, Office of Science, Office of Basic Energy Sciences, Materials Sciences and Engineering Division under Contract No. DE-AC02-05CH11231 (van der Waals heterostructures program, KCWF16). Z.Q. acknowledges the Future Materials Discovery Program through the National Research Foundation of Korea (No. 2015M3D1A1070467), Science Research Center Program through the National Research Foundation of Korea (No. 2015R1A5A1009962), King Abdullah University of Science and Technology (KAUST) under Award No. ORA-CRG10-2021-4665. C.X. acknowledges financial support from the National Key R&D Program of China (No. 2022YFA1402901), NSFC (No. 12274082), Shanghai Science and Technology Committee (Grant No. 23ZR1406600), Shanghai Pilot Program for Basic Research-FuDan University 21TQ1400100 (23TQ017). B.Z. acknowledges the support from the China Postdoctoral Science Foundation (Grant No. 2022M720816). F.M. and J.Y. acknowledge support from the U.S. Department of Energy, Office of Science, Office of Basic Energy Sciences, Materials Sciences and Engineering Division under contract DE-AC02-05-CH11231 (Organic-Inorganic Nanocomposites KC3104). L.B. thanks the support of the MURI ETHOS Grant No. W911NF-21-2-0162 from the Army Research Office (ARO), the Vannevar Bush Faculty Fellowship (VBFF) Grant No. N00014-20-1-2834 from the Department of Defense and award no. DMR-1906383 from the National Science Foundation Q-AMASE-i Program (MonArk NSF Quantum Foundry). H.P. acknowledges support from the Army Research Laboratory via the Collaborative for Hierarchical Agile and Responsive Materials (CHARM) under cooperative agreement W911NF-19-2-0119. Y.T.S. acknowledges additional financial support from the startup funding at University of Southern California. This research used resources of the Advanced Light Source, which is a DOE Office of Science User Facility under contract no. DE-AC02-05CH11231. K.X. and A.M. acknowledge cryo-EM support from the US Department of Energy, Office of Basic Energy Sciences, Division of Materials Science and Engineering under contract DE-AC02-76SF00515. P.B. and R.R. acknowledge the support of the Army Research Office under the ETHOS MURI via cooperative agreement W911NF-21-2-0162. The devices for transport measurements were fabricated in the UC Berkeley Marvell Nanofabrication Laboratory. The electron microscopy studies were performed at the Cornell Center for Materials Research, a National Science Foundation (NSF) Materials Research Science and Engineering Centers program (DMR-1719875, NSF-MRI-1429155). The microscopy work at Cornell was supported by the NSF PARADIM (DMR-2039380), with additional support from Cornell University, the Weill Institute and the Kavli Institute at Cornell. The authors acknowledge M. Thomas, J. G. Grazul, and M. Silvestry Ramos for technical support and careful maintenance of the instruments.

## Author contributions
H.Z., X.C., R.J.B. and R.R. conceived the project and designed experiments; X.C. synthesized the crystals and performed the magnetization measurements, assisted by N.G.; N.S. performed single-crystal XRD; H.Z. carried out MFM measurements and electric transport measurements; Y.T.S. and K.X. conducted the STEM measurements under the supervision of D.A.M. and A.M.; Y.T.S. performed the LTEM measurements under the supervision of D.A.M; B.Z. did the DFT calculations under the supervision of C.X. and L.B.; T.W. and H.Z. performed the XMCD and PEEM measurements assisted by Z.Q., A.S., A.N.D. and P.S.; T.W. performed the micromagnetic simulation assisted by H.Z.; F.M., X.H., Y.J. and H.Z. fabricated the devices with assistance from X.Z.C., S.H. and H.P.; P.M. implemented statistical analysis; T.Z. and Z.H. carried out the STM measurements under the supervision of M.F.C.; P.B. performed the SHG measurements under the supervision of A.R.; J.Y. and L.W.M. gave suggestions and commented on the manuscript. H.Z., X.C. and R.R. wrote the manuscript. All authors discussed the results and commented on the manuscript.

## Competing interests
The authors declare no competing interests.

## Additional information

[1]Department of Materials Science and Engineering, University of California, Berkeley, CA 94720, USA. [2]Materials Sciences Division, Lawrence Berkeley National Laboratory, Berkeley, CA, 94720, USA. [3]Mork Family Department of Chemical Engineering and Materials Science, University of Southern California, Los Angeles, CA 90089, USA. [4]School of Applied and Engineering Physics, Cornell University, Ithaca, NY 14853, USA. [5]Department of Physics, University of California, Berkeley, CA 94720, USA. [6]Center for Neutron Science and Technology, School of Physics, Sun Yat-Sen University, Guangzhou, Guangdong 510275, China. [7]Key Laboratory of Computational Physical Sciences (Ministry of Education), Institute of Computational Physical Sciences, State Key Laboratory of Surface Physics, and Department of Physics, Fudan University, Shanghai 200433, China. [8]Shanghai Qi Zhi Institute, Shanghai 200030, China. [9]Department of Mechanical Engineering, Stanford University, Stanford, CA, USA. [10]Advanced Light Source, Lawrence Berkeley National Laboratory, Berkeley, CA 94720, USA. [11]Molecular Foundry, Lawrence Berkeley National Laboratory, Berkeley, CA 94720, USA. [12]Department of Materials Science and NanoEngineering, Rice University, Houston, TX 77005, USA. [13]Department of Chemistry, Rice University, Houston, TX 77005, USA. [14]Department of Physics and Astronomy, Rice University, Houston, TX 77005, USA. [15]Rice Advanced Materials Institute, Rice University, Houston, TX 77005, USA. [16]Physics Department and Institute for Nanoscience and Engineering, University of Arkansas, Fayetteville, AR 72701, USA. [17]Kavli Institute at Cornell for Nanoscale Science, Cornell University, Ithaca, NY 14853, USA. [18]These authors contributed equally: Hongrui Zhang, Yu-Tsun Shao, Xiang Chen, Binhua Zhang. ✉e-mail: hongruizhang@berkeley.edu; chenx889@mail.sysu.edu.cn; csxu@fudan.edu.cn; rramesh@berkeley.edu

