## [Peer Review File · Nature Communications]

REVIEWER COMMENTS

Reviewer #1 (Remarks to the Author):

Please see attached file

Reviewer #2 (Remarks to the Author):

In this manuscript entitled “Spin disorder control of topological spin texture”, the authors report a new approach to control and manipulate topological solitons by injecting random spins between the vdW gap. This approach allows the authors to tune local magnetic interactions, which leads to the observations of order-disorder magnetic domain and skyrmion lattice transition in layered Fe₃GaTe₂. Also, they presented the impact of different level of intercalation along with electrical control of topological spin textures. The present study substantiates the significance of introducing disordered spins in vdW Fe₃GaTe₂ in order to control the observed spin textures for low-dimensional spintronics devices.

My detailed comments and suggestions are as follows:

1. How does the disordered spins induce a bifurcation between the zero-field cooling (ZFC) and field cooling (FC) magnetization-temperature (M-T) curves?
2. In Supplementary Fig. 3, the caption for denoting both out-of-plane and in-plane magnetic fields are written as blue, which is not matching with the fig.
3. Can you justify, why only at low temperature with lower concentration of Feint for thick Fe₃GaTe₂, the antiferromagnetic phase is stabilized?
4. The third-nearest neighbor (J_{3nn} - J_{3nn}) favors weak antiferromagnetism with $J_{3nn} = 4.3$ meV. Justify it.
5. Why does the Feint -Feint interaction favors a strong antiferromagnetic coupling, whereas the other nearest neighbor interactions dominantly prefers ferromagnetic coupling?
6. In the caption of Fig. 2, the unit for average area size of the Ferri-phase domains is wrongly written as μm^2 instead of micrometer².
7. Why does there in no direct visualization or contribution from the antiferromagnetic domains as predicted by DFT calculations to the intensity in MFM images?
8. On page no. 7, the figure numbers assign to represent “Gaussian fitting of the image intensity distribution” should be Figure 3c and 3g instead of Fig. 4c and 4g.
9. How do the authors justify the possible cause for shrinking skyrmionium as the pulse number increases?

The authors demonstrated the modulation of local anisotropic magnetic interactions in a van der Waals (vdW) ferromagnet Fe₃GaTe₂ via the interaction of magnetic iron atoms within the vdW gap. They observed the order-disorder magnetic domain and skyrmion lattices and demonstrated the electrical control of non-trivial topological solitons, e.g. skyrmioniums and skyrmion bags, at room temperature. Overall, this is a thorough study of topological spin textures in the layered magnet Fe₃GaTe₂. However, I find the current version doesn't demonstrate enough interesting/novel phenomena or significant advances in the area of magnetism and topology. The manuscript needs to specify the follow points:

1. The current version doesn't show enough novelty that warrants the publication of this paper in Nature Communications. Firstly, intercalated Fe in Fe₃GeTe₂ system has already been demonstrated [Adv. Mater. 34, 2108637 (2022)]. In current study, Fe₃GaTe₂ shares the similar crystal structure and Fe vacancies with Fe₃GeTe₂, thus it is not surprise to see the topological magnetic structures in Fe₃GaTe₂. Secondly, the manipulation of room-temperature magnetic skyrmions by current has also been widely demonstrated [e.g. Nat. Mater. 15, 501–506 (2016); Nano Lett. 17, 2703–2712(2017)]. So what's new about their study? The authors should specify the novelty or the special features of this study that can support its publication in Nature Communications.
2. The core of this draft is the spins of intercalated irons that can be antiferromagnetically coupled, giving rise to disorder and resultant exotic spin textures. However, I find the evidence of the antiferromagnetical coupling between the intercalated irons is not solid. Spin-flip transition can be solely induced by multi-domains in thick Fe₃GaTe₂ nanoflakes. This multi-domain induced anomalous Hall loops (e.g. in Fig. S4b) can be widely observed in thick nanoflakes in FeGeTe family. Also, exchange bias effect can be induced by ferromagnetic-ferrimagnetic interface, the observation of EB effect is not a sufficient condition for AFM coupling. The authors should clarify why this spin-flip transition in thick Fe₃GaTe₂ nanoflakes can be directly correlated to AFM phase.
3. After checking the Methods section, I suppose the MFM measurements were conducted in the air at room temperature, the AFM phase can also be induced by naturally oxidized layer, especially in FeGeTe family. Similarly, Fe₃GaTe₂ is also easy to be oxidized, thus it is normal to detect the AFM phase and unusual magnetic domains. The authors should conduct more experiments and exclude this possibility.
4. In Fig. S3, the MT curves indeed exhibit kinks near T_f, possibly indicating the AFM phase near T_f. However, the EB effect was observed at low temperature, far below T_f. Why the EB effect was observed at low temperature but not near T_f ?
5. In line 960, the authors claimed that “*both skyrmions and skyrmioniums are not motived by small pulse current below 5GA/m². While slightly above ~ 5 GA/m², skyrmioniums start to move and shrink and, finally, collapse into a skyrmion*”. That means skyrmioniums are useless, since they are not stable under current pulse.

6. Some of the phrases in current version are confusing and should be further polished. For example, “*When the field was swept back and forced between -0.21 T and 0.21 T*” (line 813), I suppose the phrase here should be “back and forth”.

Response to the Reviewers' Reports

We thank the reviewers for their careful reading of our manuscript and the helpful suggestions and comments. Given below are the detailed, point-by-point responses to the questions and suggestions. The newly inserted parts and the changes made in the main text and supplementary information are highlighted in green.

Response to Reviewer #1

The authors demonstrated the modulation of local anisotropic magnetic interactions in a van der Waals (vdW) ferromagnet Fe₃GaTe₂ via the interaction of magnetic iron atoms within the vdW gap. They observed the order-disorder magnetic domain and skyrmion lattices and demonstrated the electrical control of non-trivial topological solitons, e.g. skyrmioniums and skyrmion bags, at room temperature. Overall, this is a thorough study of topological spin textures in the layered magnet Fe₃GaTe₂. However, I find the current version doesn't demonstrate enough interesting/novel phenomena or significant advances in the area of magnetism and topology. The manuscript needs to specify the follow points:

Response: We greatly appreciate Reviewer #1's positive comments on our work:" Overall, this is a thorough study of topological spin textures in the layered magnet Fe₃GaTe₂." We also thank Reviewer #1 for carefully reading our manuscript and giving comments and suggestions. We believe that we have addressed all of Reviewer #1's concerns and highlighted the novelty of this work. We hope Reviewer #1 will be satisfied with the revisions to our manuscript.

Comment 1. *The current version doesn't show enough novelty that warrants the publication of this paper in Nature Communications. Firstly, intercalated Fe in Fe₃GeTe₂ system has already been demonstrated [Adv. Mater. 34, 2108637 (2022)]. In current study, Fe₃GaTe₂ shares the similar crystal structure and Fe vacancies with Fe₃GeTe₂, thus it is not surprise to see the topological magnetic structures in Fe₃GaTe₂. Secondly, the manipulation of room-temperature magnetic skyrmions by current has also been widely demonstrated [e.g. Nat. Mater. 15, 501–506 (2016); Nano Lett. 17, 2703–2712(2017)]. So what's new about their study? The authors should specify the novelty or the special features of this study that can support its publication in Nature Communications.*

Response: We thank the reviewer for the comment. In the following, we describe the novel aspects of our work:

1) In reference [Adv. Mater. 34, 2108637 (2022)], the stabilization of ferromagnetic skyrmions induced by vacancies of the Fe^{mid} atoms in Fe_3GeTe_2 was reported. However, they only demonstrated the existence of skyrmions, whereas our study reveals a much wider variety of topological spin textures. Furthermore, while they did mention the existence of intercalated iron atoms between the van der Waals gap, unfortunately, they did not discuss the role of intercalation in stabilizing skyrmions.

2) References [Nat. Mater. 15, 501–506 (2016)] and [Nano Lett. 17, 2703–2712 (2017)] are two classical works that focus on the formation of traditional skyrmions in magnetic multilayer films. They demonstrate that pulse currents can induce traditional skyrmion motion, which has been the focus of researchers in the past decade. However, what is particularly noteworthy in our work is that we demonstrated the current-induced skyrmionium motion, not the traditional skyrmion motion. It is equally important to note that our work demonstrates the difference between these two topological objects in terms of their current-induced motion.

3) The traditional skyrmion can be stabilized by competition amongst magnetic exchange energy, dipolar energy, Dzyaloshinskii–Moriya interaction, and magnetic anisotropy energy. However, tuning those above isotropic magnetic interactions can only stabilize traditional skyrmion and manipulate their size and density. We propose that by introducing local disordered magnetic interactions, we can flexibly manipulate the ordering, magnetic ground state, and topological number of the topological spin texture. Based on this, we emphasize several pioneering observations in the field of topological solutions: a) we first clearly show the ferromagnetic (disorder) and ferrimagnetic (order) skyrmions phase separation, b) we observed skyrmionium and skyrmion ‘bags’ at room temperature in 2D materials for the first time, c) we demonstrated current-induced skyrmionium motion in 2D materials for the first time and verified this motion without the skyrmion Hall effect. d) our principle will be applicable to other intercalated van der Waals magnets when locally spatial inversion symmetry is induced. We believe that this work provides a pathway for designing new topological spin textures and possibly opens the door to studying the room-temperature novel topological solutions beyond skyrmions in 2D magnets.

In summary, we believe that this work represents significant advancements and novelty beyond previous reports and, therefore, should be worthy of consideration for publication.

Comment 2. *The core of this draft is the spins of intercalated irons that can be antiferromagnetically coupled, giving rise to disorder and resultant exotic spin textures. However, I find the evidence of the antiferromagnetical coupling between the intercalated irons is not solid. Spin-flip transition can be solely induced by multi-domains in thick Fe₃GaTe₂ nanoflakes. This multi-domain induced anomalous Hall loops (e.g. in Fig. S4b) can be widely observed in thick nanoflakes in FeGeTe family. Also, exchange bias effect can be induced by ferromagnetic-ferrimagnetic interface, the observation of EB effect is not a sufficient condition for AFM coupling. The authors should clarify why this spin-flip transition in thick Fe₃GaTe₂ nanoflakes can be directly correlated to AFM phase.*

Comment 2.1 *I find the evidence of the antiferromagnetical coupling between the intercalated irons is not solid. Spin-flip transition can be solely induced by multi-domains in thick Fe₃GaTe₂ nanoflakes. This multi-domain induced anomalous Hall loops (e.g. in Fig. S4b) can be widely observed in thick nanoflakes in FeGeTe family. The authors should clarify why this spin-flip transition in thick Fe₃GaTe₂ nanoflakes can be directly correlated to AFM phase.*

Response: We thank the reviewer for the comment. The antiferromagnetic disordered spins in Fe₃GaTe₂ with Fe^{int} originate from two sources: 1) the antiferromagnetic coupling between the sublayers (A-type antiferromagnetic phase) and 2) the antiferromagnetic coupling between intercalated iron atoms. The fraction of intercalated iron atoms in the phase-separated Fe₃GaTe₂ system is only 5% ~ 8.5%. The antiferromagnetic coupling between intercalated iron atoms can induce local anisotropic magnetic interactions and subsequently increase the number of stripe dislocations in the magnetic domain pattern, potentially leading to the emergence of novel topological spin textures. However, due to the lower Fe^{int} ratio in the entire Fe₃GaTe₂ system, its influence on the domain contrast can be considered negligible. Therefore, the macroscopic magnetic behavior, such as spin-flip transition and exchange bias, is mainly determined by the existence of A-type antiferromagnetic coupling between the sublayers rather than between the intercalated iron atoms.

We fully agree with the reviewer that there are two different types of magnetic domains in Fe₃GaTe₂, which is also confirmed by our real-space observation in Fig. 2. Additionally, we also

observed two different types of magnetic domains in the similar magnet, Fe_3GeTe_2 , in Fig. S12. We have labeled the two different regions as **Ferro-phase** and **Ferri-phase**, determined by the contrast intensity as shown in Fig. 2 and Fig. S12. In all the magnetic imaging measurements, we did not observe two different domain regions with the same intensity but exhibiting distinct magnetic behaviors.

The **Ferri-phase** can be regarded as a coexistence of the A-type antiferromagnetic phase and the ferromagnetic phase. There are several reasons for the existence of the A-type antiferromagnetic phase in Fe_3GaTe_2 , described below:

1) Due to the low barrier energy between ferromagnetic and A-type antiferromagnetic state, the A-type antiferromagnetic phase is commonly observed in 2D ferromagnetic materials with strong perpendicular magnetic anisotropy, especially in thick nanoflakes/bulk, for example, CrTe_2 [Adv. Funct. Mater. 2022, 32, 2202977], Fe_3GeTe_2 [Adv. Mater. 2023, 35, 2302568, 2D Mater. 4, 011005 (2017)], etc... Electric fields [Nature Nanotech 13, 549–553 (2018)], chemical doping [Phys. Rev. Mater. 4, 074008 (2020), Phys. Rev. B 109, 104402 (2024)], and strain [Nat. Mater. 18, 1303–1308 (2019), Adv. Mater. 2023, 35, 2203411] can easily induce the transition from ferromagnetic to A-type antiferromagnetic phase. Similar to Fe_3GeTe_2 , our DFT calculations of Fe_3GaTe_2 also agrees with this conclusion (Fig. S8).

2) The Fe_3GaTe_2 system exhibits a strong perpendicular magnetic anisotropy, with its magnetic anisotropic energy even surpassing that of CoFeB thin films at room temperature. [Nat Commun 13, 5067 (2022)]. This indicates a preference for spin alignment along the c -axis. Based on the PEEM measurement and 4D-LSTEM observations, no in-plane component of spins was observed in the **Ferri-phase** regions. Therefore, the low-intensity contrast in the magnetic images can be attributed to the presence of the A-type antiferromagnetic phase but the tilted spins.

3) Both in-plane and out-of-plane magnetization decreases in Fe_3GaTe_2 with 8.5% Fe^{int} near T_f which suggests that the kink is not a spin reorientation but the emergence of an antiferromagnetic phase. In addition, the occurrence of exchange bias at low temperatures indicates the existence of the antiferromagnetic phase.

In summary, we believe that the A-type antiferromagnetic phase should exist in the thick Fe_3GaTe_2 nanoflakes.

In the manuscript, due to the weak antiferromagnetic coupling at high temperatures, the evolution of the magnetic domain as a function of the magnetic field is very complicated. In contrast, transitions observed at low temperatures and high magnetic fields in the anomalous Hall curves correspond to spin-flip transitions occurring. In Fig. R1_1, the estimated ratio of the A-type antiferromagnetic phase in the 400-nm-thick nanoflake is only $\sim 14.1\%$. Referring to Fig. R1_2a, at low temperatures, process “1” primarily denotes the switching of the majority of ferromagnetic domains. The presence of a small amount of A-type antiferromagnetic phase in Fig. R1_2a necessitates a larger magnetic field for alignment, as indicated by process “2”. The antiferromagnetic domain size ($\sim 2 \mu\text{m}^2$ for 400 nm nanoflake in Fig. 1_1b, c) decreases as the scale along the c -direction decreases. Consequently, it becomes difficult to stabilize a long-range antiferromagnetic phase in thin nanoflakes. As a result, the spin flop transition field decreases as the thickness decreases, eventually vanishing in 110 nm thick nanoflakes (Fig. R1_2c).

Fig. R1_1 The statistical size of phase domains from the large-scale MFM image. **a**, The MFM image was obtained at room temperature and zero field in a 400-nm-flake Fe_3GaTe_2 nanoflake with 8.5% Fe^{int} . The image size is $20 \times 20 \mu\text{m}^2$. **b**, The **Ferri-phase** domain areas are counted. The domain number is marked in Fig. **a**. The total area and average area of **Ferri-phase** domains are ~ 113.1 and $2.3 \pm 1.5 \mu\text{m}^2$, respectively. Combined with the quantitative 4D-LTEM results, we denote the proportions of ferromagnetic (FM) and antiferromagnetic (AFM) components by:

$$f_{AFM} = \frac{Area_{FiM}}{Area_{FM}} \times \frac{(Intensity_{FM} - Intensity_{FiM})}{Intensity_{FM}}$$

and $f_{FM} = 1 - f_{AFM}$, respectively. The estimated AFM domain ratio at room temperature is $\sim 14.1\%$.

c, Histogram of **Ferri-phase** domain areas indicates that most areas are larger than $2 \mu\text{m}^2$.

Fig. R1_2. Thickness-dependent anomalous Hall curves for Fe_3GeTe_2 nanoflake with 8.5% Fe^{int} at 20 K. $t = 350$ nm (a), 140 nm (b), and 110 nm (c)

Based on the explanation provided above, we have revised the related statement in the Supplementary Information Figure S4 to ensure clarity, where it now states:

Supplementary Fig. 4. Thickness-dependent anomalous Hall curves at various temperatures of Fe_3GaTe_2 . **a**, Isothermal magnetization curves of a bulk crystal at various temperatures. **b-e**, Anomalous Hall curves of Fe_3GaTe_2 nanoflakes with 8.5% Fe^{int} ($t = 350$ nm, 140 nm, and 110 nm)

and with 65.3% Fe^{int} ($t = 150$ nm) obtained at different temperatures. A kink was observed at high magnetic fields in the anomalous Hall curves at low temperatures in Figs. **b** and **c**, marked by the arrows, attributed to the spin-flip transition of the A-type antiferromagnetic phase. Since the antiferromagnetic domain size decreases as the scale along the c -direction decreases, it is difficult to stabilize a long-range antiferromagnetic order in thin nanoflakes, even at low temperatures. Thus, for $t = 110$ nm, the anomalous Hall curves display standard square shapes without a spin-flop transition at low temperatures in Fig. **d**. No visible spin-flip transition occurs at low temperatures in Fe₃GaTe₂ nanoflakes with 65.3% Fe^{int} (Fig. **e**), attributed to the relatively high energy barrier between the antiferromagnetic and ferromagnetic state at high Fe^{int} concentrations. Hence, the antiferromagnetic state is readily stabilized at low temperatures in thick Fe₃GaTe₂ nanoflakes with lower Fe^{int} concentrations.

Comment 2.2 Also, exchange bias effect can be induced by ferromagnetic-ferrimagnetic interface, the observation of EB effect is not a sufficient condition for AFM coupling.

Response: The reviewer is correct. The ferromagnetic-ferrimagnetic interface can induce exchange bias. We did not claim that we observed a pure antiferromagnetic phase in the manuscript. Instead, we believe that the A-type antiferromagnetic phase exists in the ferromagnetic background. Broadly speaking, the coexistence of ferromagnetic and antiferromagnetic parts can be regarded as a ferrimagnetic phase, which is also what we marked in Fig. 2.

Comment 3. After checking the Methods section, I suppose the MFM measurements were conducted in the air at room temperature, the AFM phase can also be induced by naturally oxidized layer, especially in FeGeTe family. Similarly, Fe₃GaTe₂ is also easy to be oxidized, thus it is normal to detect the AFM phase and unusual magnetic domains. The authors should conduct more experiments and exclude this possibility.

Response: We thank the reviewer for the suggestions. The following experiments can help exclude the influence of the oxidized layer.

1) Fig. 1_3 depicts the atomic force microscopy and magnetic force microscopy image of the Pt(2nm)/Ti(2nm)/Fe₃GaTe₂//SiO₂/Si samples. The surface of the sample exhibits high-quality atomic-level terraces in Fig. 1_3a, indicating no degradation on the surface. The two distinct types of magnetic domains were also clearly observed in Fig. 1_3b, suggesting that the antiferromagnetic phase is not induced by the surface oxidized layer in our case.

Fig. R1_3 a, Atomic force microscopy, and **b**, Magnetic force microscopy image of a 260-nm-thick Fe_3GaTe_2 nanoflake with 8.5% Fe^{int} . 2 nm Ti/2 nm Pt capped the nanoflake to prevent surface oxidation.

2) To further verify the surface quality, we conducted XMCD-PEEM measurements on the Fe_3GaTe_2 nanoflakes at room temperature, which is a surface-sensitive measurement (~ 5 nm). We can clearly observe the coexistence of two different contrast regions in Fig. 2i, similar to the bulk-sensitive LSTEM and MFM measurements in Fig. 2h, k. Additionally, a similar domain pattern was also observed in the XMCD-PEEM image on the Fe_3GeTe_2 nanoflake, as shown in Fig. S12. These findings indicate that the antiferromagnetic phase is also present in the non-oxidized $\text{Fe}_3\text{Ga}(\text{Ge})\text{Te}_2$ nanoflake.

3) The unusual magnetic domain in the Fe_3GaTe_2 system is more pronounced in thick nanoflakes compared to thin nanoflakes. This observation is consistent with thickness-dependent anomalous Hall measurements at low temperatures, where no spin-flop transition was detected in the thin nanoflake samples (Fig. R1_2). Typically, if a surface oxidized layer were present, its effects would be more noticeable in thin nanoflakes.

4) The domain width differs between the **Ferri-phase** and **Ferro-phase** regions, obtained by both bulk-sensitive LTEM measurement and surface-sensitive XMCD-PEEM measurement in Fig. 2i-l. Due to the weak stray field of the **Ferri-phase**, its domain width and skyrmion size are smaller than that of the **Ferro-phase**. This indicates that the antiferromagnetic layer is on the micro-scale, randomly located within the nanoflake instead of being induced on the surface by an oxidized layer.

5) Under the same measurement conditions, this unusual magnetic domain was not observed in the Fe_3GaTe_2 nanoflakes with 65.3% Fe^{int} . The intensity of the MFM image at various thicknesses of the nanoflake remains uniform in Fig. S10. This is also consistent with the conclusion of the DFT calculation in Fig. S8, which suggests that the barrier(s) energy between the A-type antiferromagnetic state and the ferromagnetic state becomes higher as the Fe^{int} ratio increases.

In summary, while the idea of an oxidized layer inducing the antiferromagnetic phase in Fe_3GaTe_2 may remain debatable [Adv. Funct. Mater. 2023, 33, 2214007, Adv. Mater. 2023, 35, 2203411], our findings suggest that the presence of the antiferromagnetic phase in our Fe_3GaTe_2 samples with Fe^{int} was not induced by the surface oxidized layer.

Comment 4. In Fig. S3, the MT curves indeed exhibit kinks near T_f , possibly indicating the AFM phase near T_f . However, the EB effect was observed at low temperature, far below T_f . Why the EB effect was observed at low temperature but not near T_f ?

Response: We thank the reviewer for the comment. In this study, we utilize the occurrence of exchange bias as one piece of evidence for the existence of the antiferromagnetic phase. We did not delve into the details of exchange bias in the Fe_3GaTe_2 system. However, in order to address the reviewer's concern clearly, we measured the exchange bias effect at room temperature, as shown in Fig. R1_4. As the reviewer expected, we definitely did not observe the measurable exchange bias effect at room temperature.

Fig. R1_4 Anomalous Hall curves obtained at room temperature following the field-cooling process. No measurable exchange bias field was observed.

In ferromagnetic/antiferromagnetic systems, the blocking temperature (T_B , defined by the threshold temperature beneath which the exchange bias shows up) can be much lower than the Néel temperature (T_N) of the antiferromagnets [PRL. 84, 6102(2000), PRL 98, 217202 (2007)]. Especially for highly disordered systems such as spin glass systems [Nat. Mater. 6, 70–75 (2007)], the antiferromagnetic spins have weak intrinsic anisotropy so as to rotate with the FM spins between T_N and T_B . In our case, the antiferromagnetic spins can be aligned by a magnetic field at room temperature (near spin-frozen temperature). This is verified by the magnetic field-dependent MFM measurements, combined with the anomalous Hall measurement in Fig. S15. In contrast, below T_B (20 K in Fig. S6d), the uncompensated antiferromagnetic spins become stable and do not rotate with ferromagnetic spins during the hysteresis loop measurements, giving rise to the exchange bias effect.

Comment 5. *In line 960, the authors claimed that “both skyrmions and skyrmioniums are not motived by small pulse current below 5GA/m². While slightly above ~ 5 GA/m², skyrmioniums start to move and shrink and, finally, collapse into a skyrmion”. That means skyrmioniums are useless, since they are not stable under current pulse.*

Response: We thank the reviewer for the comment. Unlike the skyrmion lattice phase, the stabilization energy of each skyrmion in magnetic multilayer films may be different. After applying a large pulse current, due to the spin-orbit torque or thermal effect, the phase transition from a skyrmion to a ferromagnetic phase may occur. For example, in the Pt/Co/Ta system, in the reference (Nat. Mater. 15, 501–506 (2016)), mentioned in Comment 1, most of the skyrmions marked in red dotted of Fig. R1_5 transition into the ferromagnetic phase after a pulse current.

Fig. R1_5 Comparison of skyrmion states before and after pulse current in Pt/Co/Ta system. (Nat. Mater. 15, 501–506 (2016)) The skyrmions in the red dots vanished after applying pulse currents.

In our experiment, we conducted current density-dependent skyrmionium motion measurements to investigate the threshold value current density for the topological phase transition from skyrmionium to skyrmion, as depicted in Fig. S16. From our observations, we determined that when the current density exceeds $\sim 11 \text{ GA/m}^2$, a topological phase transition occurs, resulting in the topological number transition from 0 to 1. Subsequently, in the subsequent experiment, we aimed to demonstrate current-induced skyrmionium motion while ensuring that the current density remained below $\sim 11 \text{ GA/m}^2$ to prevent the occurrence of such a topological phase transition. Consequently, we successfully realized current-induced skyrmionium motion at room temperature without observing the skyrmion Hall effect. This significant advancement paves the way for potential applications of this material in magnetic racetrack memory devices.

Comment 6. *Some of the phrases in current version are confusing and should be further polished. For example, “When the field was swept back and forced between -0.21 T and 0.21 T ” (line 813), I suppose the phrase here should be “back and forth”.*

Response: We thank the reviewer for pointing it out. We have revised it in the updated manuscript.

Response to Reviewer #2

In this manuscript entitled “Spin disorder control of topological spin texture”, the authors report a new approach to control and manipulate topological solitons by injecting random spins between the vdW gap. This approach allows the authors to tune local magnetic interactions, which leads to the observations of order-disorder magnetic domain and skyrmion lattice transition in layered Fe₃GaTe₂. Also, they presented the impact of different level of intercalation along with electrical control of topological spin textures. The present study substantiates the significance of introducing disordered spins in vdW Fe₃GaTe₂ in order to control the observed spin textures for low-dimensional spintronics devices.

Response: We sincerely appreciate Reviewer #2’s positive evaluation of our manuscript. We also thank Reviewer #2 for carefully reading the manuscript and valuable suggestions. We have revised the manuscript following the suggestions of Reviewer #2. We hope Reviewer #2 will be satisfied with the revisions to our manuscript.

Comment 1. *How does the disordered spins induce a bifurcation between the zero-field cooling (ZFC) and field cooling (FC) magnetization-temperature (M-T) curves?*

Response: We thank the reviewer for the comment. In the spin glass (-like) system, below the spin-frozen temperature (T_f), the magnetic behavior as a function of temperature becomes history-dependent due to the spins being in a frozen phase [Rev. Mod. Phys. 58, 801 (1986), arXiv:2208.00981]. During the field cooling measurement procedure, a magnetic field is applied above the Curie temperature, and magnetization is measured upon slow cooling in the field. Most of the spins align along the direction of the magnetic field, resulting in a magnetization-temperature curve resembling those observed in normal ferromagnetic behavior. In contrast, during the zero-field cooling measurement procedure, the sample is cooled without applying a magnetic field, and the field is applied at the lowest temperature. Since the spins are in a frozen state, the magnetization cannot reach an equilibrium value, as it does in the field cooling procedure. Instead, it slowly increases with time. Then, as the temperature increases, the magnetization-temperature curve progressively rises, eventually merging with the field cooling curve in the vicinity of T_f .

Comment 2. *In Supplementary Fig. 3, the caption for denoting both out-of-plane and in-plane magnetic fields are written as blue, which is not matching with the fig.*

Response: We thank the reviewer for the careful read. We have corrected this typo and highlighted it in green in the Supplementary Information. The following sentence, marked in green, is the revised caption:

The temperature-dependent magnetization (M-T) was measured under both the out-of-plane (blue) and in-plane (orange) magnetic fields...

Comment 3. *Can you justify, why only at low temperature with lower concentration of Fe^{int} for thick Fe₃GaTe₂, the antiferromagnetic phase is stabilized?*

Response: We thank the reviewer for the comment. The antiferromagnetic disordered spins in our Fe₃GaTe₂ sample with Fe^{int} originate from two sources, as determined by DFT calculations: 1) the antiferromagnetic coupling between the sublayers (referred to as the A-type antiferromagnetic phase) and 2) the antiferromagnetic coupling between intercalated irons. The antiferromagnetic coupling between intercalated irons exists in all Fe₃GaTe₂ samples with Fe^{int}, which can induce a local magnetic interaction and subsequently increase the number of stripe dislocations in the magnetic domain pattern. However, due to the lower Fe^{int} ratio in the entire Fe₃GaTe₂ system, its influence on the domain contrast can be considered negligible. Thus, from the MFM measurements, we can conclude that the A-type antiferromagnetic phase tends to be more stable in the thick Fe₃GaTe₂ nanoflakes with lower concentrations of Fe^{int}. The explanation is as follows: 1) From the DFT calculation in Fig. S8, the ferromagnetic state is always the most favored state. However, at a relatively low intercalation level, the antiferromagnetic state is also accessible. Combined with the pinning effect at low intercalation levels, as well as thermal fluctuations at higher temperatures, it becomes easier to access the antiferromagnetic phase. At much higher concentrations, because of the stronger interlayer coupling, it is easier to access the ferromagnetic phase. 2) At higher temperatures, the thermal fluctuation will cause the spins to fluctuate and favor a paramagnetic state. At low temperatures, it is easier to stabilize either the ferromagnetic or antiferromagnetic phase. 3) The estimated ratio of the A-type antiferromagnetic phase in the 400-nm-thick nanoflake is only ~ 14.1% in Fig. R2_1. The antiferromagnetic domain size (~ 2 μm² for 400 nm nanoflake in Fig. 2_1b, c) decreases as the scale along the *c*-direction decreases. Thus, it is not easy to stabilize a long-range antiferromagnetic phase in the thin nanoflake.

Fig. R2_1 The statistical size of phase domains from the large-scale MFM image. **a**, The MFM image was obtained at room temperature and zero field in a 400-nm-flake sample. The image size is $20 \times 20 \mu\text{m}^2$. **b**, The **Ferri-phase** domain areas are counted. The domain number is marked in Fig. **a**. The total area and average area of **Ferri-phase** domains are ~ 113.1 and $2.3 \pm 1.5 \mu\text{m}^2$, respectively. Combined with the quantitative 4D-LTEM results, we denote the proportions of ferromagnetic (FM) and antiferromagnetic (AFM) components by:

$$f_{AFM} = \frac{Area_{FiM}}{Area_{FM}} \times \frac{(Intensity_{FM} - Intensity_{FiM})}{Intensity_{FM}}$$

and $f_{FM} = 1 - f_{AFM}$, respectively. The estimated AFM domain ratio at room temperature is $\sim 14.1\%$.

c, Histogram of **Ferri-phase** domain areas indicates that most areas are larger than $2 \mu\text{m}^2$.

Comment 4. The third-nearest neighbor ($F_{mid} - F_{mid}$) favors weak antiferromagnetism with $J_{mm} = 4.3 \text{ meV}$. Justify it.

Response: We thank the reviewer for the comment. Here, we obtain the magnetic interaction parameters from DFT calculations using the four-state energy mapping method. The third-nearest neighbor ($\text{Fe}^{\text{mid}} - \text{Fe}^{\text{mid}}$) favors weak antiferromagnetism, which may result from the competition between the direct exchange between $\text{Fe}^{\text{mid}} - \text{Fe}^{\text{mid}}$ sites and the superexchange mediated through the Ga ions. It is similar to the isomorphic Fe_3GeTe_2 , which also has been reported as weak antiferromagnetism between $\text{Fe}^{\text{mid}} - \text{Fe}^{\text{mid}}$ in previous work (Adv. Mater. **2022**, 34, 2107779).

Comment 5. *Why does the Fe^{int}-Fe^{int} interaction favors a strong antiferromagnetic coupling, whereas the other nearest neighbor interactions dominantly prefers ferromagnetic coupling?*

Response: We thank the reviewer for the comment. For the Fe₃GaTe₂ with Fe^{int} system, the magnetic coupling is governed by the direct exchange interaction and the superexchange interaction. 1) For the direct exchange interaction, the *d* orbitals in the nearest-neighbor Fe atoms overlap directly without a mediation atom. Thus, it gives rise to antiferromagnetic coupling. 2) For the superexchange interaction, the *d* orbitals on the nearest-neighbor Fe atoms overlap with the *p* orbitals of Ga or Te atoms, giving rise to ferromagnetic /antiferromagnetic coupling sensitive to the Fe-Te (Ga)-Fe bond angles. Note that the superexchange ferromagnetic/antiferromagnetic coupling becomes weaker/stronger when the Fe-Te (Ga)-Fe bond angles are increased from 90° to 180° (according to Goodenough–Kanamori–Anderson rules, 90° for ferromagnetic coupling and 180° for antiferromagnetic coupling).

In our present case, the Fe^{int}-Te-Fe^{int} bond angles (~110°) are larger than Fe^{top}-Te-Fe^{top} bond angles (~90°), thus yielding a weaker superexchange ferromagnetic coupling than the latter. As a result, for Fe^{int}-Feⁱⁿ, the direct exchange interaction plays the dominant role, giving rise to an antiferromagnetic coupling. For the Fe^{top}-Fe^{top}, the superexchange ferromagnetic interaction is stronger than the direct exchange interaction, leading to a ferromagnetic coupling.

Comment 6. *In the caption of Fig. 2, the unit for average area size of the Ferri-phase domains is wrongly written as um2 instead of micrometer2.*

Response: We thank the reviewer for the careful read. We have corrected this typo and highlighted it in green in the main text.

Comment 7. *Why does there in no direct visualization or contribution from the antiferromagnetic domains as predicted by DFT calculations to the intensity in MFM images?*

Response: We thank the reviewer for the comment. Monte Carlo simulations were employed to confirm that the coexistence of antiferromagnetic and ferromagnetic coupling is achievable. Considering that the simulated material scale on DFT calculation is much smaller than the actual experimental scale. It is challenging to match experimental and theoretical results rigorously.

Comment 8. On page no. 7, the figure numbers assign to represent “Gaussian fitting of the image intensity distribution” should be Figure 3c and 3g instead of Fig. 4c and 4g.

Response: We thank the reviewer for the detailed read. We have corrected this typo and highlighted it in green in the main text.

Comment 9. How do the authors justify the possible cause for shrinking skyrmionium as the pulse number increases?

Response: The reviewer raises an interesting question. The shrinking of Skyrmionium in Fe_3GaTe_2 could possibly be caused by the current-induced out-of-plane damping-like torque. Although Fe_3GaTe_2 does not exhibit global inversion symmetry breaking, as evidenced by its space group- $P6_3/mmc$, determined by single-crystal XRD measurements (Fig. S1), locally both in-plane and out-of-plane mirror symmetry breaking may still exist due to iron atom vacancies or intercalations. When pulsed currents are applied to the nanoflake, a spin current with out-of-plane spin polarization is produced, resulting in an out-of-plane spin-orbit torque. This torque switches the magnetization of Fe_3GaTe_2 , leading to a ferromagnetic phase. Consequently, the pulsed current reduces the length of the ring-shaped magnetic domain (skyrmionium), making it approach the final ferromagnetic state. Another possible reason is the thermal effects induced by the pulse current. The skyrmion size and domain width decrease with increasing temperature [Nat Commun 13, 3035 (2022)]. The skyrmionium can be viewed as composed of a skyrmion with topological charge number (Q) = +1 and a skyrmion with Q = -1, resulting in Q = 0. Therefore, the size of the skyrmionium also decreases with increasing temperature. To verify this, we performed temperature-dependent MFM measurements, as shown in Fig. R2_2, where the skyrmionium size indeed decreased with increasing temperature.

Fig. R2_2 The evolution of skyrmionium as the temperature increases in the Fe_3GeTe_2 nanoflakes with 8.5% Fe^{int} are depicted in the images **(a)** and **(b)**, corresponding to nanoflake thicknesses of 119 nm and 400 nm, respectively. It is evident that the size of the skyrmionium decreases as the temperature increases.

REVIEWERS' COMMENTS

Reviewer #1 (Remarks to the Author):

The authors have addressed all my concerns. This study demonstrated the current-induced skyrmionium motion, the first demonstration of the motion of non-trivial topological solitons beyond traditional skyrmions should deserve its publication. Thus I recommend the publication of this revised version in Nature Communications.

Reviewer #2 (Remarks to the Author):

The authors have addressed all the queries and concerns of the reviewers satisfactorily. The manuscript is also revised as per suggestions. The revised manuscript may be considered for publication in the journal.

Reviewer #1 (Remarks to the Author):

The authors have addressed all my concerns. This study demonstrated the current-induced skyrmionium motion, the first demonstration of the motion of non-trivial topological solitons beyond traditional skyrmions should deserve its publication. Thus I recommend the publication of this revised version in Nature Communications.

Response: We thank the reviewer for the positive recommendation.

Reviewer #2 (Remarks to the Author):

The authors have addressed all the queries and concerns of the reviewers satisfactorily. The manuscript is also revised as per suggestions. The revised manuscript may be considered for publication in the journal.

Response: We thank the reviewer for the positive recommendation.